# *Mycobacterium tuberculosis* associated with severe tuberculosis evades cytosolic surveillance systems and modulates IL-1β production

Jeremy Sousa[1,2,3,17], Baltazar Cá [1,2,3,17], Ana Raquel Maceiras [1,2], Luisa Simões-Costa[1,2], Kaori L. Fonseca[1,2,3], Ana Isabel Fernandes[1,2], Angélica Ramos [4], Teresa Carvalho[4], Leandro Barros [1,2], Carlos Magalhães[5,6], Álvaro Chiner-Oms[7], Henrique Machado [5,6], Maria Isabel Veiga[5,6], Albel Singh[8], Rui Pereira[1,9], António Amorim [1,9,10], Jorge Vieira[1,2], Cristina P. Vieira[1,2], Apoorva Bhatt[8], Fernando Rodrigues[5,6], Pedro N. S. Rodrigues[1,2,11], Sebastien Gagneux[12,13], António Gil Castro[5,6], João Tiago Guimarães [4,14], Helder Novais Bastos[1,2,14,15], Nuno S. Osório [5,6], Iñaki Comas[7,16] & Margarida Saraiva[1,2✉]

Genetic diversity of *Mycobacterium tuberculosis* affects immune responses and clinical outcomes of tuberculosis (TB). However, how bacterial diversity orchestrates immune responses to direct distinct TB severities is unknown. Here we study 681 patients with pulmonary TB and show that *M. tuberculosis* isolates from cases with mild disease consistently induce robust cytokine responses in macrophages across multiple donors. By contrast, bacteria from patients with severe TB do not do so. Secretion of IL-1β is a good surrogate of the differences observed, and thus to classify strains as probable drivers of different TB severities. Furthermore, we demonstrate that *M. tuberculosis* isolates that induce low levels of IL-1β production can evade macrophage cytosolic surveillance systems, including cGAS and the inflammasome. Isolates exhibiting this evasion strategy carry candidate mutations, generating sigA recognition boxes or affecting components of the ESX-1 secretion system. Therefore, we provide evidence that *M. tuberculosis* strains manipulate host-pathogen interactions to drive variable TB severities.

---

[1] i3S-Instituto de Investigação e Inovação em Saúde, University of Porto, Porto, Portugal. [2] IBMC-Instituto de Biologia Molecular e Celular, University of Porto, Porto, Portugal. [3] Doctoral Program in Molecular and Cell Biology, ICBAS-Instituto de Ciências Biomédicas Abel Salazar, University of Porto, Porto, Portugal. [4] São João Hospital Center & EPIUnit-Institute of Public Health, University of Porto, Porto, Portugal. [5] Life and Health Sciences Research Institute, University of Minho, Braga, Portugal. [6] ICVS/3B's-PT Government Associate Laboratory, Braga/Guimarães, Portugal. [7] Biomedicine Institute of Valencia (CSIC), Valencia, Spain. [8] School of Biosciences and Institute of Microbiology and Infection, University of Birmingham, Birmingham, UK. [9] IPATIMUP-Institute of Molecular Pathology and Immunology of the University of Porto, University of Porto, Porto, Portugal. [10] Faculty of Sciences, University of Porto, Porto, Portugal. [11] ICBAS-Instituto de Ciências Biomédicas Abel Salazar, University of Porto, Porto, Portugal. [12] Swiss Tropical and Public Health Institute, Basel, Switzerland. [13] University of Basel, Basel, Switzerland. [14] Faculty of Medicine, University of Porto, Porto, Portugal. [15] São João Hospital Center, Porto, Portugal. [16] CIBER in Epidemiology and Public Health (CIBERESP), Madrid, Spain. [17] These authors contributed equally: Jeremy Sousa, Baltazar Cá ✉email: margarida.saraiva@ibmc.up.pt

**M**ycobacterium tuberculosis is estimated to infect a quarter of the human population[1] and to kill >1.5 million people every year[2]. During its parallel evolution with the human host[3], M. tuberculosis developed important immune evasion mechanisms, including virulence factors aimed at preventing elimination by macrophages[4], and strategies to modulate T-cell responses to favor transmission[5]. M. tuberculosis is an obligate human pathogen with no environmental reservoir, and for which transmission relies on disease establishment[6]. Thus, M. tuberculosis must create a balance between damaging the host (virulence) and finding the opportunity to spread (transmission)—a balance which is ultimately achieved by modulating the host immune response. Interestingly, some strains of M. tuberculosis are more transmissible than others, and this transmission potential varies in different human genetic backgrounds[7]. Therefore, one can expect a relevant role for both host and pathogen diversity in disease establishment and transmission, through the modulation of host immune responses.

The human adapted tuberculosis (TB)-causing bacteria are part of the Mycobacterium tuberculosis complex (MTBC), and can be divided into seven distinct lineages that exhibit a strong phylogeographical structure[6]. Despite harboring little DNA sequence variation as compared to other bacteria[8], strains of the MTBC differ in their capacity to modulate the immune response[9]. Pathogen diversity within the MTBC also impacts the clinical manifestation of TB[7,9]. What remains unknown, however, is the interaction between pathogen-induced immune-modulation and disease severity. In other words, how does the natural diversity in M. tuberculosis isolates direct the host immune response towards a certain TB presentation. We studied well-defined patient and pathogen populations to disclose the relevant immune responses leading the various disease outcomes. Through genomic, transcriptional, and functional analyses we propose that phylogenetically related M. tuberculosis isolated from severe TB cases develops mechanisms to escape cytosolic recognition and consequently lower cytokine production by host cells. This study contributes to our understanding of the modulation of host immunity to TB, with the potential to inform the design of host-directed and pathogen-directed therapies for this devastating disease.

## Results

**TB cohort characterization**. To investigate whether M. tuberculosis-associated determinants may contribute to the pathogenesis of TB through the manipulation of host immune responses, we started by stratifying TB severity in a fully characterized population of TB patients[10]. From a cohort of 813 culture-positive TB cases[10], 681 adult pulmonary TB patients were selected (Supplementary Fig. 1a) and classified according to the absence or presence of risk factors for TB (Fig. 1a). To exclude the effect of host-related comorbidities on TB presentation, we next focused on TB only patients. The genetic ancestry of 60 TB patients (Supplementary Fig. 1a) was investigated through a validated panel of autosomal ancestry-informative markers (AIMs). This analysis was performed alongside reference genomes of African, European, East Asian and Native American, as well as a group of Portuguese reference genomes[11,12] and a group of TB contacts from Porto (Fig. 1b). The ancestry of both the Porto-TB contacts and the Porto-TB cohort seemed homogeneous, rooted on Portuguese and European makeups. Next, based on chest X-ray classification and systemic parameters (Fig. 1c and Supplementary Fig. 1b[13]), a clinical decision system to assign the severity of TB at presentation was developed and applied to the 133 TB cases, for which all required data was available (Supplementary Fig. 1a). The most common TB

presentation was moderate, followed by mild and severe (Fig. 1d). Although a higher proportion of men was observed in the severe TB group, both the age of the patient and reported time of symptoms were similarly distributed within the three TB severity groups Supplementary Fig. 1c). In parallel, we recovered the infecting M. tuberculosis isolates (Supplementary Fig. 1a) and investigated the genetic structure of the bacteria population. A large predominance of the MTBC L4 and of sublineage L4.3/LAM was revealed within the pathogen population (Fig. 1e), in line with other reports focused in Europe[14,15]. The pathogen population structure was replicated across the different TB severity groups (Fig. 1f).

**Severe TB-associated M. tuberculosis induce lower cytokine response**. Our findings suggested that host-pathogen sympatry breaks do not likely explain differences in the severity of TB in the cohort under study. We then questioned whether M. tuberculosis isolated from patients presenting different TB severities interacted differently with human immune cells. For this, we selected L4 isolates from the mild (n = 10), moderate (n = 9), or severe (n = 7) TB groups (Supplementary Table 1). Each of these isolates was used to infect peripheral blood mononuclear cells (PBMC) from unrelated donors. To control for the TB status of these control donors, we accessed their interferon gamma release assay (IGRA) results. This assay discriminates individuals who present T-cell memory responses to M. tuberculosis (and are IGRA +) from those who do not (and are IGRA–). For our assays, we used PBMCs isolated from 8–10 donors who had been IGRA + for >2 years, did not undergo preventive antibiotherapy and had no past TB episode. We decided not to use cells from active TB patients, as their cytokine responses are altered by the disease status. We also did not use IGRA– donors, as within this population certain individuals might present innate resistance to TB and altered myeloid cell responses to infection[16]. The supernatants from the infected cultures were collected 24 h post-infection and the amounts of IL-12p40, IL-1β, IFN-β, and IL-10 quantified. Interestingly, as compared to M. tuberculosis isolated from mild TB cases, M. tuberculosis isolates associated with severe or moderate forms of TB induced lower cytokine responses in infected PBMCs (Fig. 1g). IFN-β secretion was below detection level for all cases. Although PBMCs from different donors varied in their absolute amount of cytokine production, the observed cytokine pattern was primarily controlled by M. tuberculosis (Supplementary Fig. 2). These data suggest a pathogen-driven association between the levels of cytokines triggered and TB severity.

**M. tuberculosis SNPs associated with host responses**. To search for bacterial genomic determinants driving the differential cytokine responses in PBMCs, the whole-genome sequence of the 26 M. tuberculosis clinical isolates was generated. A total of 3112 unique polymorphisms (SNPs and InDels) were found in comparison with the inferred sequence of the MTBC ancestor[5]. The maximum pairwise distance between any two clinical isolates was 456 mutations, as expected for M. tuberculosis strains from the same lineage[9]. Five possible transmission clusters were identified (Supplementary Fig. 3a). Taking IL-1β as a proxy for low versus high-cytokine responses and using 3077 genome-wide SNP positions, no clear associations between phylogeny and IL-1β induction were found (Fig. 2a). Unbiased comparative analysis between the selected M. tuberculosis isolates revealed 21 genes harboring polymorphisms shared exclusively by >5 low IL-1β-inducing isolates and three genes harboring polymorphisms shared exclusively by three high inducing isolates (Fig. 2b and Supplementary Table 2).

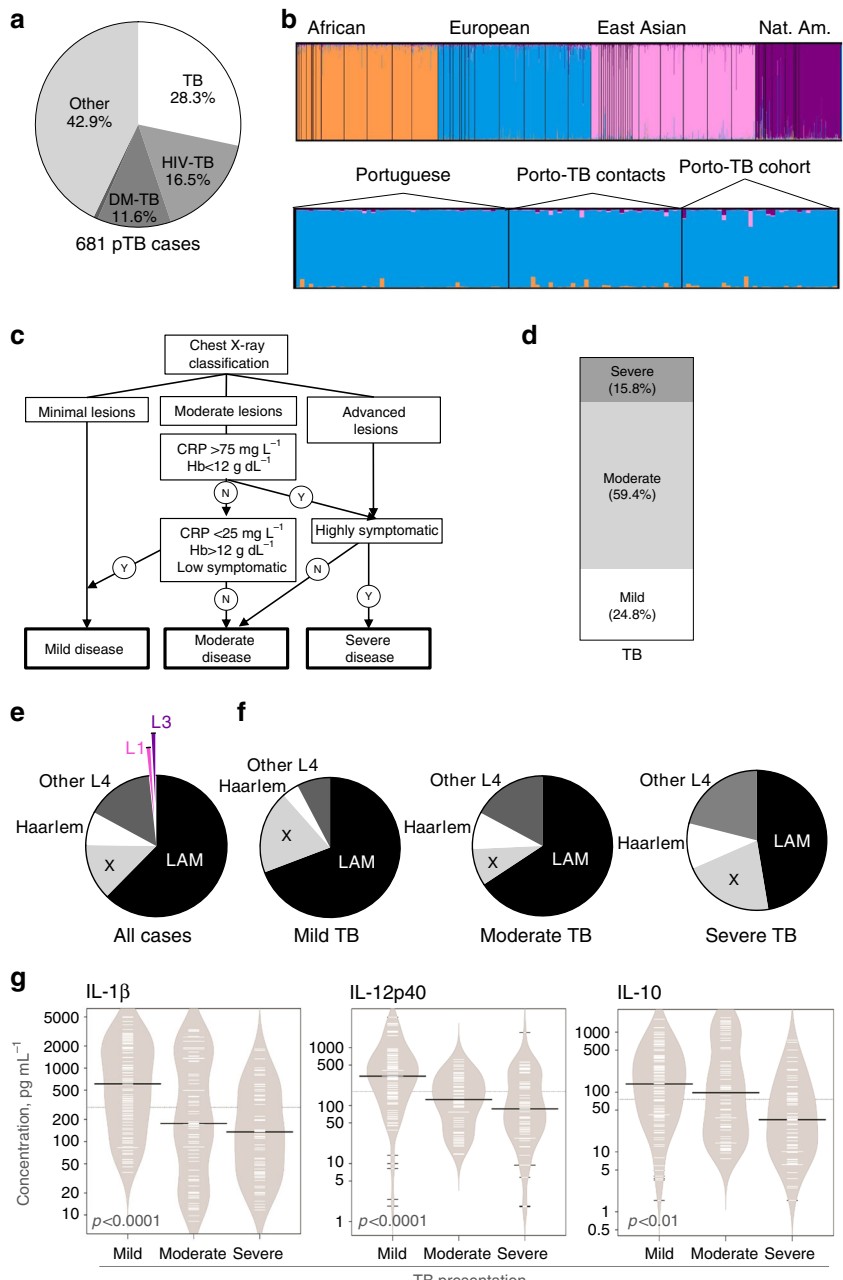

**Fig. 1 M. tuberculosis clinical isolates associated with severe TB induce lower cytokine responses. a** The clinical data for 681 adult pulmonary TB (pTB) cases were reviewed and classified according to the absence or presence of comorbidities. **b** Top panel: genetic makeup through a validated panel of autosomal ancestry-informative markers of reference populations of African, European, East Asian, and Native American biogeographical origin. Bottom panel: the genetic ancestry for a group of 60 TB patients was determined, together with TB contacts from the same area and a reference Portuguese population. Represented are European ancestry in blue, African in orange, East Asian in pink and Native American in purple. **c** Clinical decision system developed to classify the severity of TB at presentation in mild, moderate or severe. **d** Patients who had no known comorbidities ($n = 133$) were classified into mild, moderate, or severe TB cases at presentation. **e** *M. tuberculosis* isolates recovered for 117 of the 133 selected TB patients were classified into lineages and those belonging to lineage 4 (L4) were further classified into L4.3/LAM, L4.1.1/X, L4.1.2/Haarlem or other lineage 4 sublineages. **f** Mild, moderate, and severe TB cases, as determined in **d**, were stratified according to the sublineage, the infecting bacteria belonged to. Statistical analysis was performed with Fisher's exact test. **g** PBMC from 8-10 IGRA + donors were infected with L4 *M. tuberculosis* isolated from mild ($n = 10$), moderate ($n = 9$), or severe ($n = 7$) TB patients. An MOI of 1 was used and 24 h post-infection the amounts of secreted IL-1β, IL-12p40, IL-10, and IFN-β quantified. Represented are bean plots for each cytokine. Black lines show the medians; white lines represent individual data points; polygons represent the estimated density of the data. IFN-β was undetected for all conditions. Statistical analysis was performed with Pearson correlation; R and *p*-values were: IL-1β $R = -0.3471$; R2 $= 0.1205$; $p < 0.00001$; IL-12 $R = -0.3419$; R2 $= 0.1169$; $p < 0.00001$; and IL-10 $R = -0.1918$; R2 $= 0.0368$; $p = 0.002473$. CRP C-reactive protein, Hb hemoglobin, LAM Latin-American and Mediterranean, N no, TB tuberculosis, Y yes.

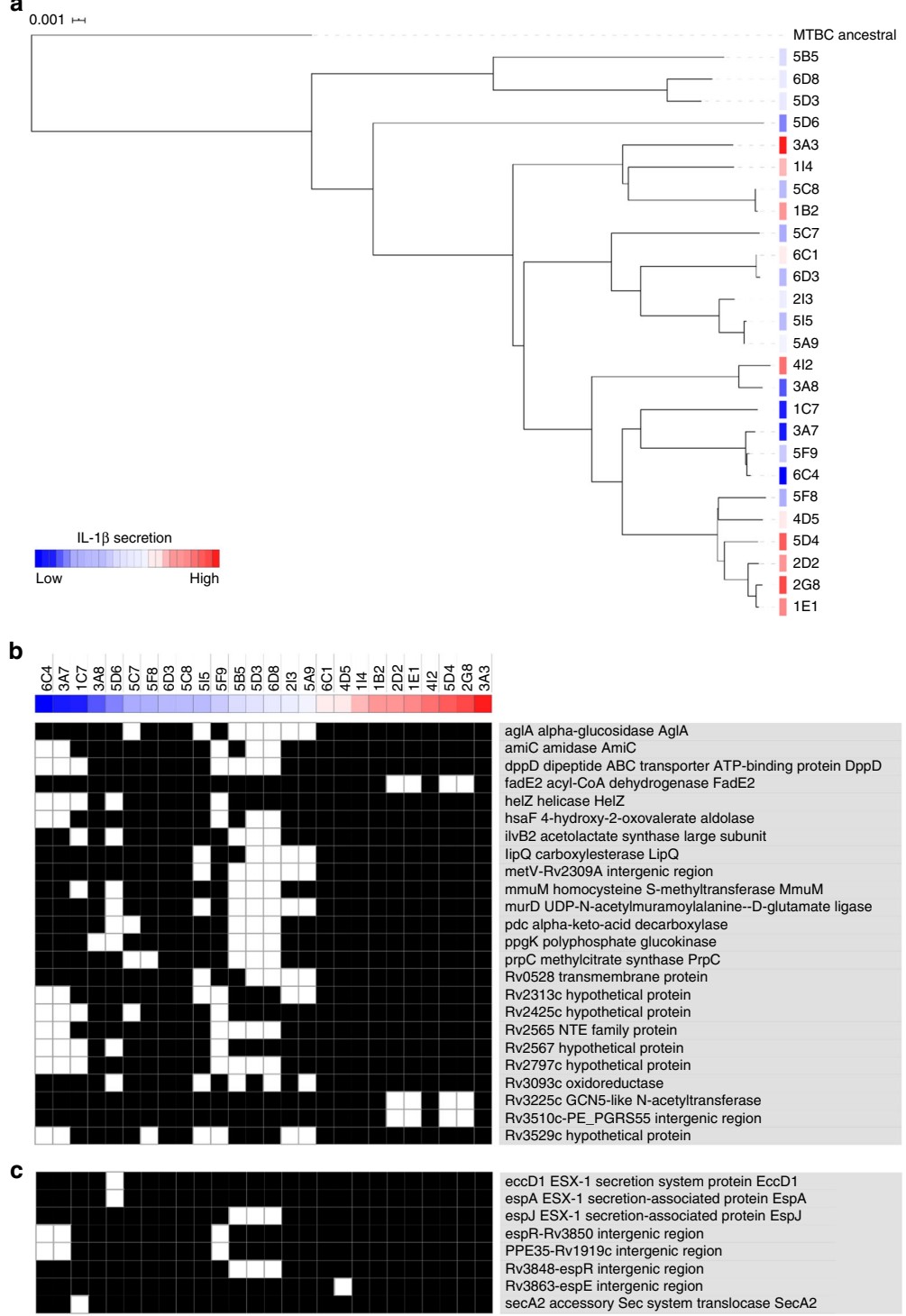

**Fig. 2 Common polymorphisms in groups of low-cytokine-inducing *M. tuberculosis* clinical isolates.** Genome-wide analysis was performed for 26 *M. tuberculosis* clinical isolates with known IL-1β inducing profiles in human PBMCs. **a** Phylogenetic tree of the isolates constructed with genome-wide SNPs and using maximum likelihood. The color gradient between dark blue (low) and dark red (high) indicates the level of IL-1β induced by each strain. **b** Genes with (white) or without (black) polymorphisms exclusively present in >30% of the *M. tuberculosis* isolates classified as low or high IL-1β inducers. **c** ESX-1 secretion system and associated regulators showing polymorphisms exclusively present in the low or high IL-1β inducing strains.

If some of the identified polymorphisms indeed lead to a modulation of cytokine induction, we would expect signatures of selection acting upon them. Homoplastic positions, those appearing multiple times and independently in the phylogeny, are generally good surrogates of the action of positive selection[17,18]. By comparing the identified polymorphisms to a collection of 4528 strain genomes representative of the global diversity of the MTBC, we found that five SNP positions from our Porto dataset also emerged in unrelated strains in the phylogeny more than one time. Notably, we found a synonymous position that occurred five times in strains from different lineages (Supplementary Table 2). Although unlikely to alter the structure

of the underlying protein, this synonymous mutation generates a new internal TANNT *sigA* recognition box. We have shown before that MTBC uses newly generated TANNT boxes as a way to adapt to changing environments[19]. In fact, among the synonymous SNPs observed, three generate new regulatory TANNT boxes (Supplementary Table 2). Taken together, the evolutionary analysis suggests that at least part of the polymorphisms and genes identified are under selection in epidemiological settings.

Given the previous association of the ESX-1 secretion system with cytokine responses, particularly IL-1β[20], a directed analysis to identify polymorphisms in this pathway was performed. We found common genetic variants shared by 5 clinical isolates (6C4, 3A7, 1C7, 5F9, and 5D6) inducing low IL-1β levels, while absent in high IL-1β inducers (Fig. 2c and Supplementary Table 3). In particular, *M. tuberculosis* isolate 5D6 harbored two SNPs in genes encoding ESX-1 secretion system components (*EccD1* and *EspA*) and isolates 6C4, 3A7, and 5F9 shared two SNPs in the intergenic regions espR-Rv3850 and PPE35-Rv1919c (Fig. 2c). No SNPs exclusive of either group of isolates were identified in the other ESX-1 components or regulators encoding genes. Collectively, our analyses support that low IL-1β induction results from different evolutionary events and highlight the diversity of bacterial genetic pathways possibly underlying the manipulation of host responses towards low-cytokine production.

## IL-1β modulation by *M. tuberculosis* is host independent.
*M. tuberculosis* isolates 6C4, 3A7, and 5F9 belong to a probable transmission cluster (Supplementary Fig. 3a) and consistently induce the secretion of low IL-1β by infecting cells, thus suggesting a "stable phenotype". Isolate 6C4 was selected as a representative of this transmission cluster and of the low-cytokine/severe TB group and *M. tuberculosis* 4I2 as a representative of the high-cytokine/mild TB. The clinical and genetic differences between these two isolates are listed in Supplementary Tables 1 and 4. Both isolates belong to the LAM sublineage of L4 and showed similar growth curves in axenic media (Supplementary Fig. 3b). We next measured the production of IL-1β induced upon infection of cells from different origins with *M. tuberculosis* isolate 4I2 or 6C4. Differential IL-1β induction was observed upon infection of PBMCs from IGRA+ (Fig. 3a) or IGRA– (Fig. 3b) donors, and upon infection of purified monocytes from IGRA– individuals (Fig. 3c), macrophages differentiated from the human monocytic cell line THP-1 (Fig. 3d) or mouse bone marrow-derived macrophages (BMDMs; Fig. 3e). The differential pattern of IL-1β expression is thus independent of the host and its TB status, being instead strongly dictated by the infecting bacteria. Of note, the differential response observed was also independent of the batch of *M. tuberculosis*, as stocks grown independently yielded the same profile of IL-1β induction in infected cells.

## *M. tuberculosis* strain-dependent transcriptome modulation.
To gain insights on the mechanisms underlying the differential cellular response to the selected *M. tuberculosis* isolates, a targeted RNA Sequencing for gene expression analysis[21,22] was obtained for BMDMs before and after infection with either isolate. An overall similarity between replicates and groups was detected for genes with a >2-log fold-change and a statistical difference (adjusted *p*-values of ≤0.05) in expression upon infection (Supplementary Fig. 4a, b). Principal component analysis (PCA) showed that over 75% of the differences observed between samples, described by PC1, are mainly between BMDMs left uninfected and BMDMs infected with the 4I2 *M. tuberculosis* isolate (Fig. 4a). In response to infection with *M. tuberculosis* isolate 4I2,

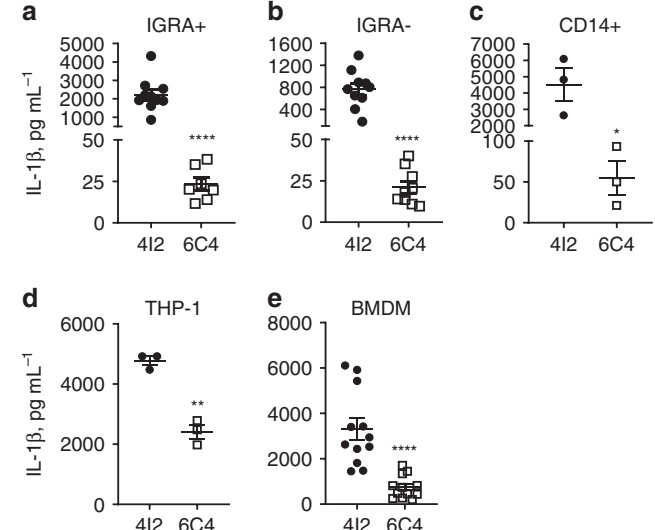

**Fig. 3 *M. tuberculosis* manipulate IL-1β induction independent of host TB status.** *M. tuberculosis* isolates 4I2 and 6C4 were selected based on the TB presentation, MTBC sublineage and intensity of cytokine induction in infected PBMCs. PBMCs from IGRA + (**a**) or IGRA– (**b**) donors; CD14 + monocytes purified from IGRA– donors (**c**); PMA-differentiated THP-1 cells (**d**) or C57BL/6 mouse BMDMs (**e**) were infected with *M. tuberculosis* isolate 4I2 (solid circles or bars) or 6C4 (open squares or bars) for 24 h and the amount of secreted IL-1β quantified in the culture supernatants by immunoassay. MOIs of 1 (**a–d**) or 2 (**e**) were used for infection. Represented is the mean ± SEM; *n* = 10 donors (**a**, **b**), or *n* = 3 donors (**c**) or *n* = 3 wells from one (**d**) or four (**e**) independent experiments. Undetected values are not represented. Statistical analysis was performed using two-tailed unpaired Student's *t*-test (**p* < 0.05; ***p* < 0.01; ****p* < 0.001; and *****p* < 0.0001).

1.271 transcripts were differentially expressed by BMDM, against only 389 transcripts detected in the case of isolate 6C4 (Fig. 4b, c). Canonical pathway analysis (ReactomePA) revealed that whereas infection with isolate 4I2 activated PRR-related pathways, such as TLRs, NLRs, and MyD88 or TRIF cascades, infection with 6C4 did not (Fig. 4d, e). An overview of the interactions between the activated pathways and the significantly altered genes in each pathway upon infection is shown in Supplementary Fig. 4c, d. Thus, infection with *M. tuberculosis* isolate 4I2 resulted in a more robust and complex alteration of the gene expression profile of host BMDMs than that induced by isolate 6C4.

## Increased IL-1β secretion is due to inflammasome activation.
Differentially regulated genes within the cytokine/interleukin-related pathways showed an overall higher expression in BMDMs infected with *M. tuberculosis* isolate 4I2 (Fig. 5a). However, the expression of *Il1b* was detected at the same level in BMDMs infected with either isolate (Fig. 5a). This was validated by real-time PCR in both BMDMs (Fig. 5b) and human CD14 + monocytes (Supplementary Fig. 5a). Furthermore, for both *M. tuberculosis* isolates the production of IL-1β by infected BMDMs was fully dependent on TLR2 activation (Fig. 5c), with no contribution of TLR4 triggering (Supplementary Fig. 5b). Therefore, the differences on secreted IL-1β by infected BMDMs likely result from post-transcriptional events, perhaps through differential inflammasome activation. To test this hypothesis, we used THP-1-ASC-GFP reporter cells[23]. The percentage of ASC-speck-positive cells in PMA-differentiated reporter cells was higher in the case of *M. tuberculosis* 4I2 infections (Fig. 5d and Supplementary Fig. 5c). Chemical inhibition of NLRP3 (Fig. 5e) or of

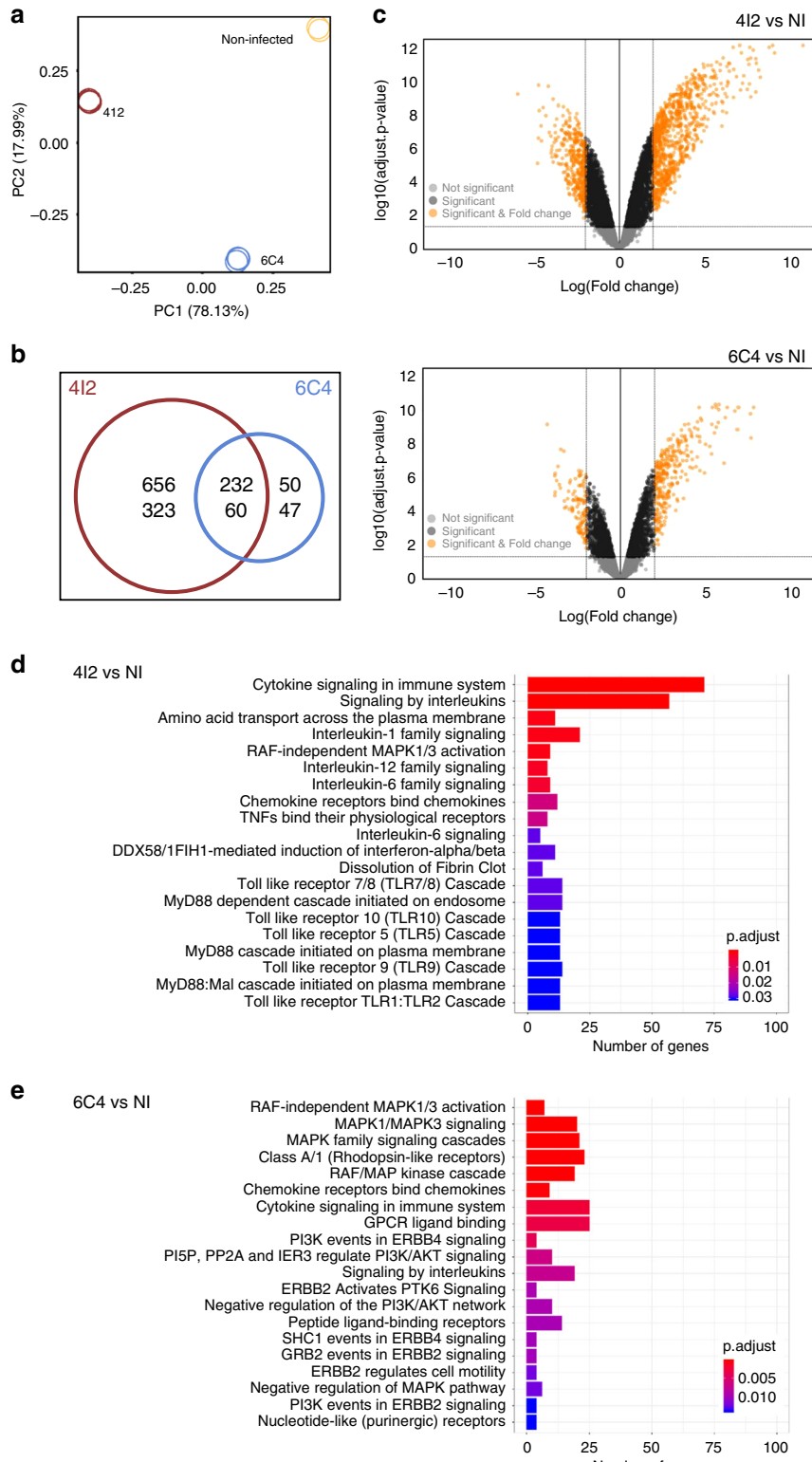

caspase-1 (Fig. 5f) strongly abrogated IL-1β production by BMDMs infected with *M. tuberculosis* isolate 4I2. Inhibition of NLRP3 or caspase-1 during infection of macrophages with the severe TB causing isolate (6C4) also decreased IL-1β secretion (Fig. 5g, h). Therefore, the activation of NLRP3 is critical for the secretion of IL-1β by BMDMs infected with either *M. tuberculosis* isolate, but isolate 4I2 shows an enhanced ability to activate the inflammasome.

To unveil the bacterial determinants of differential inflammasome activation, we firstly addressed possible differences in the cell-wall lipidic fraction. The genomic data indicated that strain 6C4 harbored SNPs in genes encoding two key *M. tuberculosis* lipids involved in virulence: sulfolipid-I and phthiocerol dimycocerosate (PDIM), suggesting a potential change in the profiles of these lipids. Lipid extracts of either bacteria were prepared and analyzed by thin-layer chromatography (TLC). No major

**Fig. 4 *M. tuberculosis* isolate associated with severe TB modulates the macrophage transcriptome.** BMDMs generated from C57BL/6 mice were infected with *M. tuberculosis* isolates 4I2 or 6C4 at a MOI of 2, or kept non-infected (NI). Six hours post-infection cell cultures were lysed, RNA extracted and subjected to targeted RNA-Seq. **a** Principal component (PC) analysis for data obtained from *M. tuberculosis* 4I2-infected (red), *M. tuberculosis* 6C4-infected (blue) and non-infected (NI, yellow) cells. Venn diagram (**b**) and volcano plots (**c**) representing the significantly differentially expressed genes (adjusted *p*-value ≤ 0.05 and log(fold-change) ≥2 or ≤ –2) when comparing BMDMs infected with *M. tuberculosis* isolate 4I2 or 6C4 with NI cells. In **c**, significantly differentially expressed genes (NI versus infected cells; adjusted *p*-value ≤ 0.05 and log(fold-change) ≥2 or ≤ –2) are shown in orange. Significant genes (adjusted *p*-value ≤0.05 and log(fold-change) between 2 and –2) are shown in black. Non-significant genes are displayed in light gray. Pathway analysis for significantly expressed genes detected in BMDMs infected with *M. tuberculosis* isolate 4I2 (**d**) or 6C4 (**e**) versus NI cells. Bar plots represent the number of genes present in each identified pathway and the color codes reflect adjusted *p*-values. In both differential expression analysis and pathway analysis, multiple testing adjustments were performed using the Benjamini–Hochberg (BH) procedure. The 20 most affected pathways are represented for each condition.

differences were observed in the lipid profiles of the two isolates, with exception of free mycolic acids that appeared in relatively higher amounts in *M. tuberculosis* isolate 4I2 (Supplementary Fig. 5d). To investigate if differences in the amount of cell-wall components might explain the differential IL-1β response by infected cells, we infected macrophages with increasing doses of *M. tuberculosis* isolate 6C4. Even when a multiplicity of infection (MOI) of 10 was used, the secretion of IL-1β was still low (Fig. 5i), whereas that of TNF increased, showing that the macrophages were upregulating their response to infection with higher doses of *M. tuberculosis* 6C4 (Supplementary Fig. 5e).

To assess for a possible active inhibition of the inflammasome and IL-1β production in cells infected with *M. tuberculosis* isolate 6C4, we next performed mixed infections of BMDMs. In these infections, the total MOI was maintained at 2, but the amount of *M. tuberculosis* isolate 4I2 was progressively replaced by isolate 6C4. Independently of the amount of isolate 6C4 present in culture, the response of BMDMs to *M. tuberculosis* isolate 4I2 was not inhibited (Fig. 5j). Furthermore, no inhibition was seen in BMDMs infected with isolate 4I2 in the presence of supernatant from cells infected with isolate 6C4 (Supplementary Fig. 5f). Infection of BMDMs with *M. tuberculosis* isolate 6C4 did not alter the transcription of key inflammasome components, as revealed through the RNA-Seq data (Supplementary Fig. 5g). Finally, infection of BMDMs with isolate 6C4 in the presence of an inflammasome activator (as ATP), an upregulation of IL-1β was observed (Fig. 5k). Altogether, this set of experiments indicate that an intrinsic property of *M. tuberculosis* isolate 6C4 underlies the failure of this isolate to activate the inflammasome, which is not related to the amount of bacteria required nor to a direct inhibition of this pathway.

***M. tuberculosis* 6C4 evades cytosolic surveillance pathways**. IL-1β-inducing *M. tuberculosis* isolates accumulated mutations in the ESX components or regulators (Fig. 2c). A competent ESX system is required for the activation of the inflammasome[20,24,25], cGAS[26–28], and RIG-I[29] in *M. tuberculosis*-infected macrophages. Thus, we next interrogated the global transcriptomic data for differences in the activation of hallmark intracellular surveillance pathways in macrophages infected with either *M. tuberculosis* isolate. A higher upregulation of the genes associated with cGAS (Fig. 6a), RIG-I (Supplementary Fig. 6a), TLR3 (Supplementary Fig. 6b) or TLR7/8/9 (Supplementary Fig. 6c) pathways was observed in BMDMs infected with *M. tuberculosis* isolate 4I2. Hierarchical clustering for cGAS and RIG-I pathways showed higher similarity between BMDMs infected with *M. tuberculosis* isolate 6C4 and non-infected cells, than between the two infections (Fig. 6a and Supplementary Fig. 6a). The enhanced transcription of the cGAS-associated genes *Ifnβ*, *Il10*, *Ccl2*, and *Cxcl10* in BMDMs infected with *M. tuberculosis* isolate 4I2 was validated by real-time PCR (Fig. 6b) and found to depend on the activation

of cGAS (Fig. 6c). Chemical inhibition of cGAS led to lesser alterations on *Tnf* transcription (Fig. 6c).

The cGAS pathway is activated by bacterial DNA, which is also a ligand for AIM2, a member of the inflammasome family known to contribute to the secretion of IL-1β[26–28]. Therefore, it is possible that the activation of AIM2 in BMDMs infected by *M. tuberculosis* isolate 4I2 is contributing to maximal IL-1β secretion. AIM2-/- BMDMs were generated and infected with either isolate. The impact of AIM2 deficiency in the secretion of IL-1β by macrophages infected with isolate 4I2 was more pronounced than that seen in the case of infection with 6C4 (Fig. 6d). Phagosome rupture by mycobacteria triggers several other molecular signals perceived by the NLRP3 inflammasome, among which K + efflux and cathepsin B release[30,31]. In further support of a more active or efficient phagosome rupture induced by *M. tuberculosis* isolate 4I2, chemical inhibition of K + channels or of cathepsin B decreased the secretion of IL-1β by 4I2-infected macrophages, while not affecting 6C4 infections (Fig. 6e, f). Blocking bacterial phagocytosis using cytochalasin D significantly compromised the secretion of IL-1β by BMDMs infected with either *M. tuberculosis* isolate (Fig. 6g), showing that intracellularly localized bacteria are required for IL-1β secretion. Of note, the phagocytosis of the two *M. tuberculosis* isolates by BMDMs occurred in a similar way (Supplementary Fig. 6d), demonstrating that differential IL-1β induction did not result from differences in bacterial internalization. Finally, rifampicin treatment of *M. tuberculosis* isolate 4I2 decreased the secretion of IL-1β by BMDMs, but had no effect in the case of 6C4 isolate (Fig. 6h). Rifampicin treatment of *M. tuberculosis* isolate 4I2 decreased the transcription of the *Ifnb* gene (Fig. 6i). Therefore, internalization of live, transcriptionally competent bacteria is required to induce maximal induction of IL-1β and IFN-β.

The differential activation of the inflammasome and cGAS by *M. tuberculosis* 4I2 versus 6C4 leads to the differential secretion of IL-1β and IFN-β two cytokines involved in cross-regulatory mechanisms[32]. To investigate whether this cross-regulation was in place independently of the *M. tuberculosis* isolate, we compared the response of IFNAR-/- BMDMs to that of WT cells upon infection with isolate 4I2 or 6C4. Absence of IFNAR further increased IL-1β secreted by BMDMs infected with isolate 4I2, but did not impact IL-1β secretion upon infection with 6C4 (Supplementary Fig. 6e). However, enhancing IFN-β production by activating BMDMs infected with *M. tuberculosis* isolate 6C4 with a RIG-I agonist decreased the secretion of IL-1β by the infected cells (Supplementary Fig. 6f). Therefore, although the mechanisms regulating the secretion of IL-1β and IFN-β are still coupled, in certain situations they may depend on the presence of other cellular sources of IFN-β to occur.

Collectively, our data suggest that *M. tuberculosis* isolates differentially manipulate phagosome rupture and the activation of several intracellular pattern recognition receptors, which

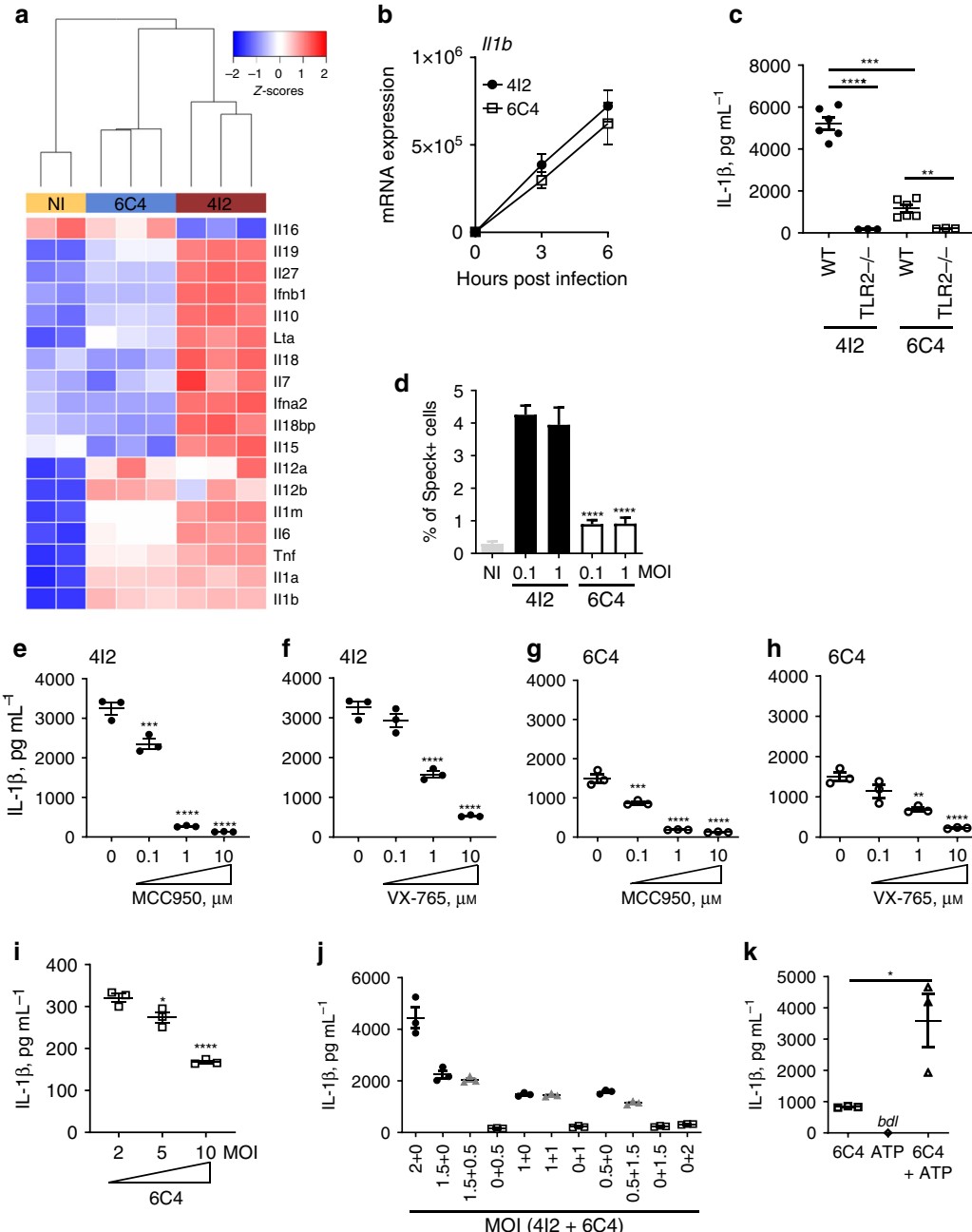

**Fig. 5 Induction of high IL-1β by *M. tuberculosis* 4I2 results from enhanced inflammasome activation. a** Gene expression heatmap representing cytokine expression for non-infected BMDMs (NI, yellow) and BMDMs infected with *M. tuberculosis* isolate 4I2 (dark red) or *M. tuberculosis* isolate 6C4 (blue). z-scores are color coded as shown. **b** C57BL/6 WT BMDMs were generated and infected with *M. tuberculosis* isolates 4I2 (solid circles) or 6C4 (open squares), and at the indicated time points, the cell cultures were lysed, RNA extracted, converted to cDNA and subjected to real-time PCR. **c** C57BL/6 WT or TLR2 deficient BMDMs were generated and infected with *M. tuberculosis* isolates 4I2 (solid circles) or 6C4 (open squares). Twenty-four hours post-infection the culture supernatants were harvested and the amount of IL-1β quantified by ELISA. **d** THP-1-ASC-GFP monocytes were PMA-differentiated and left uninfected (NI) or infected with *M. tuberculosis* isolates 4I2 (solid bars) or 6C4 (open bars) at different MOI, as indicated. Six hours post-infection, cells were fixed and stained with DAPI prior to image acquisition. The percentage of Speck-positive cells was determined for the different conditions. C57BL/6 WT BMDMs were generated and infected with *M. tuberculosis* isolates 4I2 (**e**, **f**) or 6C4 (**g**, **h**) in the absence (0) or presence of increasing doses of NLRP3 (MCC950; **e**, **g**) or Caspase-1 (VX-765; **f**, **h**) inhibitors. BMDMs were infected with increasing MOI of *M. tuberculosis* isolate 6C4 (I), with a combination of both isolates at different doses (**j**) or with *M. tuberculosis* isolate 6C4 in the absence (6C4) or presence (6C4 + ATP) of 5 mM of ATP (**k**). **e**–**k** Twenty-four hours post-infection the culture supernatants were harvested and the amount of IL-1β quantified by ELISA. Unless indicated, an MOI of 2 was used for infections. Represented is the mean ± SEM of triplicate wells for one experiment representative of two (**b**–**j**); or for one experiment (**k**). Statistical analysis was performed using two-tailed unpaired Student's t-test (**c**, **k**) or One-way ANOVA (**d**–**i**) (* $p < 0.05$; ** $p < 0.01$; *** $p < 0.001$; and **** $p < 0.0001$).

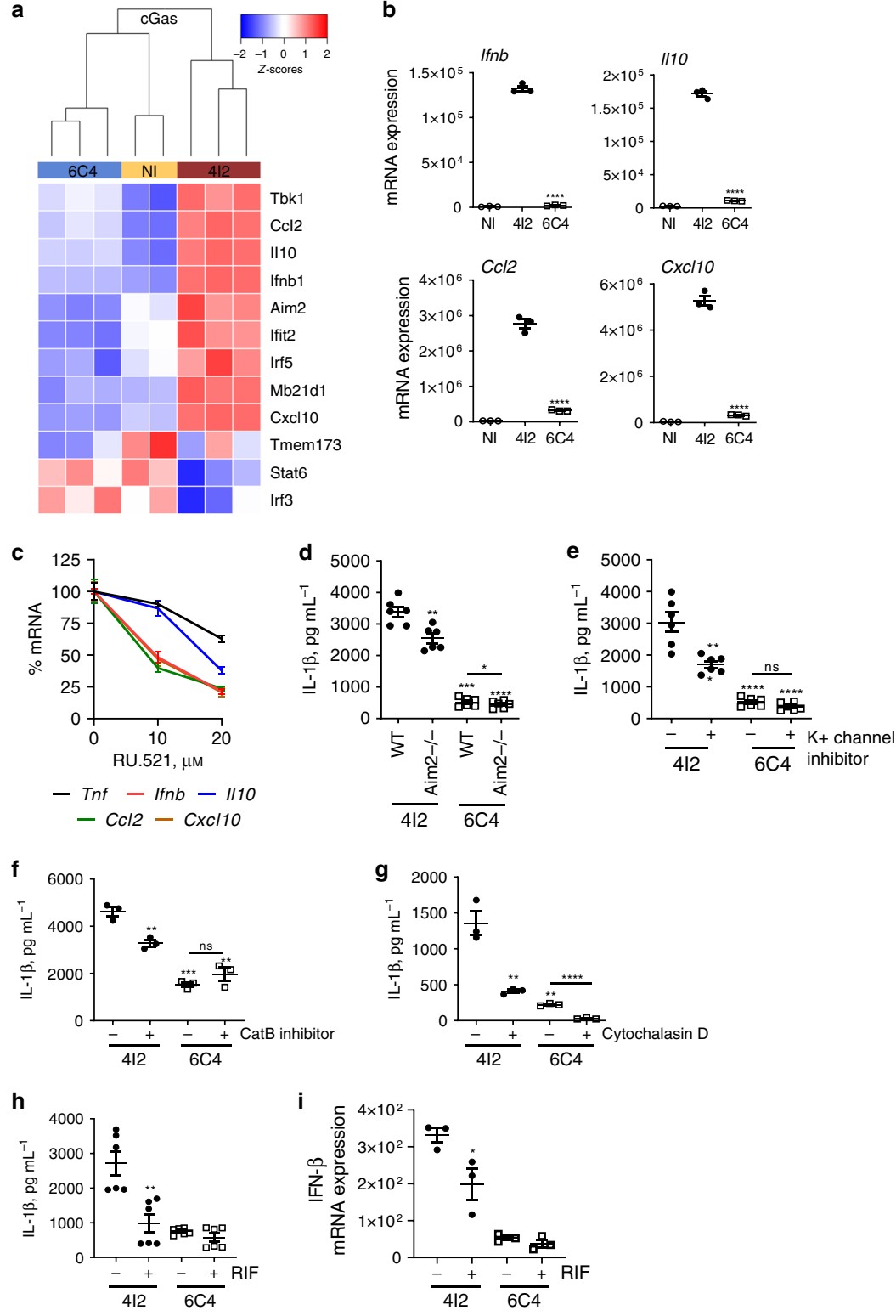

preferentially occurs in the context of a mild TB-associated *M. tuberculosis* isolate.

## Discussion

*M. tuberculosis* has been traditionally considered a clone, but evidence indicates genetic diversity within TB-causing bacteria[6]. Here, we provide insights into the functional relevance of *M. tuberculosis* diversity, linking variations in the infecting bacteria to the modulation of host immune responses and clinical outcomes of TB. Remarkably, across a range of different host cells, *M. tuberculosis* isolates recovered from severe TB cases induced lower cytokine responses as compared to those recovered from mild TB patients. This finding raises the hypothesis that *M. tuberculosis* evolved to fine-tune the immune response, ultimately modulating the pathogenesis of TB. Therefore, the reason why some individuals develop severe TB, while others do not, may also be explained by the characteristics of the infecting bacteria. Consequently, understanding the underlying bacteria molecular

**Fig. 6 M. tuberculosis isolate 6C4 evades macrophage cytosolic surveillance pathways. a** Heatmap representing the expression of genes associated with the cGAS pathway in non-infected BMDMs (NI; yellow) and BMDMs infected with *M. tuberculosis* isolate 4I2 (dark red) or *M. tuberculosis* isolate 6C4 (blue). z-scores are color coded as shown. C57BL/6 WT BMDMs were generated and infected with **b** *M. tuberculosis* isolates 4I2 (solid bars) or 6C4 (open bars) or **c** with *M. tuberculosis* isolate 4I2 in the absence (0) or presence of different doses of the cGAS inhibitor RU521. Six hours post-infection, the cell cultures were lysed, RNA extracted, converted to cDNA, and subjected to real-time PCR to quantify the expression of the indicated genes. **d** C57BL/6 WT or AIM2 deficient (-/-) BMDMs were generated and infected with *M. tuberculosis* isolates 4I2 (solid bars) or 6C4 (open bars). Infection of C57BL/6 WT BMDMs in the absence (-) or presence (+) of (**e**) a K$^+$ channel inhibitor (glybenclamide; 25 µg mL$^{-1}$), **f** a cathepsin B (CatB) inhibitor (ZRLR; 10 µM) or **g** a phagocytosis inhibitor (cytochalasin D; 5 µM). **h–i** C57BL/6 WT BMDMs were generated and infected with live or rifampicin (RIF)-treated *M. tuberculosis* isolates 4I2 (solid bars) or 6C4 (open bars). **d–h** Twenty-four hours post-infection, the culture supernatants were harvested and the amount of IL-1β quantified by ELISA. **i** Six hours post-infection the cell cultures were lysed, RNA extracted, converted to cDNA, and subjected to real-time PCR to quantify the expression of *Ifnb*. Represented is the mean ± SEM of triplicate wells from one experiment representative of two (**b**, **c**, **g**); from two (**d**, **e**, **h**) or one (**f**, **i**) independent experiments. **c** Represented is the % of inhibition for the expression of each gene in BMDMs infected with *M. tuberculosis* isolate 4I2 in the presence of RU.521 relatively to those infected in the absence of the cGAS inhibitor. An MOI of 2 was used for all infections. Statistical analysis was performed using two-tailed unpaired Student's *t*-test (*$p < 0.05$; **$p < 0.01$; ***$p < 0.001$; and ****$p < 0.0001$).

bases will prove important from a clinical standpoint. So far, we did not identify a SNP or group of SNPs associated with the immunological phenotype or the TB severity at the genome-wide level, suggesting that the ability of closely related *M. tuberculosis* to modulate the host response results from diverse events. Reflecting that variety of pathways, we have identified signatures of positive selection when comparing to a large global strain database. We have also detected an accumulation of SNPs in genes encoding components or regulators of the ESX-I secretion system in isolates associated with low IL-1β induction and high TB severity. This may hence be a hot-spot pathway explored by the bacteria to thwart full macrophage activation. It will be interesting to address this hypothesis by systematically generate *M. tuberculosis* isogenic strains containing the various SNPs in ESX-I identified here and probe their functional role in phagosome rupture and in modulating evasion to macrophage intracellular recognition pathways, as well as their role during in vivo infection, namely for the development of lung pathology. Previous studies show that lack of the ESX-I secretion system is characteristic of avirulent Mycobacteria and that abrogation of this system renders *M. tuberculosis* avirulent[33]. We now disclose further layers of complexity to this aspect of *M. tuberculosis* biology, showing that variations in the efficiency within ESX-I are likely to exist in highly related isolates, with profound implications for the pattern recognition pathways triggered in the infected macrophage and possibly for the outcome of disease. By restricting phagosome rupture and export of bacterial material to the cell cytosol, low-cytokine inducing/severe TB *M. tuberculosis* isolates prevent the activation of the inflammasome, cGAS, RIG-I, and other intracellular signaling cascades as revealed by our global transcriptomic analysis. This is in turn reflected in the lower induction of IL-1β and type I IFN.

Decreased IL-1β responses are compatible with more severe TB outcomes. Deficiencies in IL-1β production are associated with increased susceptibility of TB, both in humans and in mouse models. A multitude of mechanisms explain the protective role of IL-1 in TB, from enhancing the macrophage control of intracellular bacterial replication, to driving ILC3 responses required for early protection[34], orchestrating the relocalization of alveolar macrophages to lung interstitium[35] and limiting the proportion of infected cells and dissemination[36]. It is therefore not surprising that a professional pathogen such as *M. tuberculosis* manipulates the IL-1 pathway to delay protective mechanisms, and possibly promote dissemination and/or more severe lung lesions. Interestingly, the patient infected with the low IL-1β-inducing *M. tuberculosis* isolate 6C4 had extensive lung pathology, which included pleural involvement, thus suggesting initial dissemination events. Of note, a recent study has shown that modern strains of L4 induced higher IL-1β than ancient strains of lineages

1 and 5, and that this high IL-1β correlated with enhanced autophagy of infected macrophages[37]. Importantly, this enhanced IL-1β/autophagy axis did not result in bacterial control at the macrophage level[37]. It will be interesting to, based on our data and clinical isolates, investigate if in vivo similar mechanisms operate and what their consequences for the course of infection might be. Excessive IL-1β production may also be detrimental to the host, associating with more severe TB disease and increased lung damage[38–42]. From the pathogen's perspective, *M. tuberculosis* isolates that trigger high IL-1β responses would face the risk of being eliminated during early stages of infection, thus failing later transmission. It is thus tempting to speculate that the host-*M. tuberculosis* parallel evolution has perfected a mechanism that simultaneously controls IL-1β and its regulator IFN-β, which ultimately benefits both the host by keeping the infection and associated immune pathology at check, and the pathogen, by ensuring transmission.

Collectively, our study bears important consequences for future TB research. Host genetic or secondary immune suppression have been associated with higher TB susceptibility. We now show that the initial immune response is also modulated by differences in closely related *M. tuberculosis*. This finding calls for the inclusion of *M. tuberculosis* genetic diversity and continuous evolution as part of human genetic association studies, as well as of vaccine design and as an important component to consider in the development of host-directed therapies, such as those that interfere with the levels of IL-1β or other cytokines.

## Methods

**Ethics statement for Human sample collection**. The study protocol leading to PBMC isolation was approved by the Health Ethics Committees of the CHSJ (approval number 109-11), the North Health Region Administration (approval number 71-2014) and the Portuguese Data Protection Authority (approval number 12174-2011). To ensure confidentiality, each case was anonymized by the assignment of a random identification number. Participants were enrolled at a TB clinic in Porto and provided informed consent. Experiments were conducted according to the principles expressed in the Declaration of Helsinki.

**Animals**. Eight-to-12-week-old male or female mice used to generate BMDMs were of the following origins: C57BL/6 WT, TLR2-/- and TLR4-/- mice were maintained and provided by the animal facility of i3S; IFNAR-/- and matched C57BL/6 WT were kindly provided by Dr. Jocelyne Demangeau, IGC, Oeiras, Portugal; AIM2-/- and matched C57BL/6 WT femurs were kindly provided by Dr. Thomas Henry, CIRI-Center International de Recherche en Infectiologie (INSERM), Lyon, under the MTA OM-217356. Animals were kept under specific-pathogen free conditions, with controlled temperature (20–24 °C), humidity (45–65%), light cycle (12 h light/dark) and ad libitum food and water. All experiments were performed in strict accordance with recommendation of European Union Directive 2010/63/EU and previously approved by Portuguese National Authority for Animal Health—Direção Geral de Alimentação e Veterinária (DGAV).

**Study population**. A cohort of 813 culture-confirmed TB cases diagnosed at a University-affiliated hospital (CHSJ, Porto) during a 7 years (2007–2013) period was reviewed to derive a study group of pulmonary TB cases (Supplementary Fig. 4a). The overall demographic and clinical features of this population have been described[10]. Patients in the study group were divided into four main groups: no known comorbidities (TB group); HIV-coinfected (HIV-TB group); diabetics (DM-TB group); other comorbidities or immunological suppression, including hepatitis C virus chronic infection, alcohol abuse, end-stage chronic kidney failure, malignancy, cirrhosis or chronic liver failure, heart failure, chronic obstructive pulmonary disease, and patients with structural lung disease, such as silicosis, fibrosis, or bronchiectasis (other).

**Clinical severity classification**. The clinical records of the patients in the TB group were retrospectively reviewed to stratify the severity of TB at presentation. First, digital images of plain chest radiographs were blind-graded by two independent clinicians, using a previously published decision tree (Supplementary Fig. 1b[13]), based on the lesions extent, the presence of lung cavitation and pleural effusion. Minimal lesions were cases of single hilar enlargement, or minimal pulmonary infiltrates. Moderate disease had cavitations up to 4 cm, or pleural effusion extending to less than within 2 cm of carina, or lesions of slight to moderate density of total extent not greater than total equivalent volume of one lung, or dense and confluent lesions occupying less than one-third of one lung. Advanced disease exceeds criteria for moderate. Disagreements between the two clinicians were resolved through a consensus assessment by a third reader. To take a broader view of the disease clinical severity, a classification flowchart was developed (Fig. 1c) where chest X-ray findings were combined with measures of inflammatory consumption and overall nutritional status (baseline levels of hemoglobin and C-reactive protein), as well as severity of symptoms.

**Human ancestry**. Individual ancestry estimates were computed using Structure[41] considering four ancestral contributors (African, European, East Asian and Native American biogeographical origins; $K = 4$); known genotypes from the HGDP-CEPH panel[11] and the 1000 Genomes[42], complemented with Colombian[43–45] and Peruvian Amerindians (unpublished data), were used as learning reference samples to perform unsupervised clustering analyses of Portuguese controls (Porto TB-contacts) and TB patients (Porto TB-cohort) (100,000 burnin followed by 100,000 MCMC repetitions; three replicate runs). Aligned averaged clustering was then obtained with CLUMPP[44] and plotted with DISTRUCT[45].

**Recovery of *M. tuberculosis* clinical isolates**. Bacterial samples (n = 133) of the subjects in the TB group were recovered from stored primary cultures of *M. tuberculosis* clinical isolates at the Clinical Microbiology department of CHSJ. Two hundred mmicroliters of inoculum were plated and smeared uniformly on solid Mycobacteria 7H11 agar supplemented with 10% Oleic Albumin Dextrose Catalase Growth Supplement (OADC), 0,5% glycerol, and PANTA antibiotic mixture. The plates were incubated at 37 °C for 4 to 8 weeks. Grown colonies were gently rubbed and transferred to 20 mL of Middlebrook 7H9 liquid medium (BD Biosciences, San Jose, USA) supplemented with 10% OADC, 0,2% glycerol and 0.05% Tween® 80 (Sigma-Aldrich, St. Louis, USA). Alternatively, the stored primary cultures were re-grown in MGIT tubes using a BACTEC instrument (BD) and once a positive signal was obtained, transferred to 20 mL of Middlebrook 7H9 liquid medium, as stated above. All Middlebrook 7H9 liquid cultures were incubated at 37 °C with constant 120 rpm shaking for an additional 7–10 days, to increase the bacterial biomass.

**Preparation of bacterial stocks**. Twenty-six selected *M. tuberculosis* clinical isolates (Supplementary Table 1) were grown in 200 mL of Middlebrook 7H9 liquid medium supplemented with 10% OADC and 0,2% glycerol. Bacterial suspensions were aliquoted in cryovials, frozen, and stored at –80 °C. Bacterial quantification was performed by thawing three to five vials and plating several serial dilutions in 7H11 agar medium supplemented with 10% OADC and 0.5% glycerol. The plates were incubated 21 to 28 days at 37 °C before colony count.

***M. tuberculosis* genomic DNA extraction**. Bacterial suspensions were pelleted, resuspended in water to a final volume of 1 mL and inactivated with 0.5 mL of a 1:2 phenol-water solution for DNA extraction and homogenized with zirconia beads in TEN buffer, in the Fast-Prep24 bead beater (MP Biomedicals) at 4 M s⁻¹ for 30 s, twice. DNA was then extracted with chlorophorm, precipitated with absolute ethanol and sodium acetate, resuspended in TE buffer, quantified by spectrophotometry (NanoDrop® 1000, Thermo Scientific, Wilmington, USA) and normalized to a standard concentration of 200 ng μL⁻¹.

**MTBC lineage and sublineage genotyping**. MTBC lineage and sublineage genotyping was performed by a custom TaqMan® real-time PCR assay (Applied Biosystems, Carlsbad, USA), using SNPs as stable genetic markers, as previously described[46]. Results were analyzed with the Bio-Rad CFX Manager™ 3.1 and genotypes determined using the Allele Discrimination tool.

**Whole-genome sequencing and analysis**. DNA sequencing of bacterial isolates was performed with Illumina MiSeq and HiSeq 2000/2500 paired-end technology. Raw reads were subjected to quality trimming with Trimmomatic[47] v0.38 (minimum read length 20 and average base quality 20 in 4-base sliding windows). Visual inspection after trimming was performed with FastQC (https://www.bioinformatics.babraham.ac.uk/projects/fastqc) v0.11.7 and MultiQC[48] v1.0. Filtered reads were mapped to the reconstructed MTBC ancestral strain[5] with bwa-mem[49] and converted to BAM files with SAMtools[50] v.1.3.1. Duplicated reads were removed with Picard (http://broadinstitute.github.io/picard) Mark Duplicates v2.18.14. Single nucleotide polymorphisms and insertions/deletions were called with Pilon[51] v1.22 (minimum depth of five valid read pairs, minimum mapping quality 20 and minimum base quality 20) and filtered with bcftools (http://samtools.github.io/bcftools) v1.3.1 (keeping only biallelic sites with variant frequency >75% and with a minimum distance from indels of four nucleotides). Genomic variants were annotated with SnpEff[52] v4.3t according to the *M. tuberculosis* H37Rv reference genome annotation (NC_000962.3; GCF_000195955.2). To minimize false positives, previously identified repetitive genomic regions, mobile elements and genes containing ≥50 bp nucleotide chunks identical to other parts of the genome[53,54] were not considered to further analysis. Variants exclusive from either low or high IL-1β-inducing strains were assigned only on sites that had the reference allele successfully called on all the isolates belonging to the opposite phenotype.

Phylogenetic reconstruction was derived from a multiple sequence alignment containing only single nucleotide polymorphisms ($n = 10,748$) and allowing <5% of strains per site with missing information. Maximum likelihood (ML) tree was obtained with RaxML[55] v7.2.8 using GTR GAMMA model with estimate of proportion of invariable sites, 1000 rapid bootstrap replicates and searching for the best-scoring ML tree. Tree was rooted on the MTBC ancestor and annotated using iTOL[56] v4.3.3.

The global reference dataset was constructed with samples from[3,15,19,57–63]. We applied the pipeline explained above to derive an alignment and a phylogeny with all the downloaded samples ($n = 9240$). After that, we run Treemer[64] to reduce the number of strains with a minimum loss of genetic diversity (0.05 of the initial diversity). The final alignment consisted in 4528 MTBC strains. Variants identified in the Porto dataset were mapped to the phylogeny and the number of occurrences (homoplasies) across the phylogeny was recorded.

**Cell wall analysis**. *M. tuberculosis* isolates 4I2 and 6C4 were grown in Middlebrook 7H9 liquid medium complemented with 10% OADC and 0.2% glycerol until a dense bacteria suspension was obtained. The bacteria were pelleted by centrifugation and inactivated by autoclave sterilization before lipid extraction. Polar and apolar lipids were extracted using described methods[65]. Extracted lipids were resuspended in chloroform:methanol (2:1). Free and apolar lipid extracts using a solvent systems consisting of chloroform/methanol/water (60:16:2). A 2D TLC system B (direction 1: Petroleum ether 60–80/acetone (92:8)–3 runs; direction 2: toluene/acetone (95:5)–1 run) and system C (direction 1: chloroform/methanol (96:4)–1 run; direction 2: toluene/acetone (80:20)–1 run) were also used. Ten percent phosphomolybdic acid staining followed by charring was used to reveal lipids in TLCs.

**PBMC isolation and CD14 + cell purification**. Blood from donors was collected and processed within 2 h. PBMCs were separated using Histopaque 1077 at room temperature (Sigma-Aldrich) and following the protocol for mononuclear cell separation of SepMate-50 tubes (StemCells). Cells were resuspended in complete RPMI-1640 medium (10% Fetal bovine serum, 1% sodium pyruvate, 1% HEPES and 1% L-glutamine; cRPMI) (GIBCO), counted and plated in 24-well plates at a concentration of $0.5 \times 10^6$ cell mL⁻¹ in 500 μl of cRPMI.

CD14 + cells were isolated from cryopreserved PBMCs. PBMCs were magnetic labeled with CD14 MicroBeads (Miltenyi Biotec) and isolated through positive selection using MACS Separation Columns MS (Miltenyi Biotec), according to the manufacturer's protocol. Purified cells (~95% pure) were counted and plated in 96 well plates at a concentration of $0.5 \times 10^6$ cell mL⁻¹ in 200 μl of cRPMI (GIBCO).

**THP-1-ASC-GFP culture**. The THP-1-ASC-GFP (InvivoGen) cell line was maintained in cRPMI (GIBCO) supplemented with 100 μg mL⁻¹ Zeocin (InvivoGen), at 37 °C and 5% CO₂. Cells were used before reaching 20 passages.

**BMDM generation**. Protocols were as previously published[66]. Briefly, mice tibias and femurs were collected and bone marrow flushed with complete DMEM medium (10% Fetal bovine serum, 1% sodium pyruvate, 1% HEPES and 1% L-glutamine; cDMEM) (GIBCO). Cells were counted and plated in Petri dishes at the concentration of $0.5 \times 10^6$ cells mL⁻¹ in 8 mL with 20% L-cell conditioned medium (V/V). On day 4, cDMEM supplemented with 20% LCCM was added to the cells. On day 7, cells were harvested, counted, plated in 24-well plates at a concentration of $1 \times 10^6$ cell mL⁻¹ in 500 μl of cDMEM.

**In vitro infections**. Before infection, mycobacterial clumps were disaggregated by gentle passaging through a 25G needle. For PBMC infection, a MOI of 1 bacteria: 1 cell (MOI of 1) was used for infection. THP-1-ASC-GFP cells were primed with

100 nM PMA for 24 h, and allowed to rest for 4 days in fresh medium without PMA, before being infected. MOIs of 1 or 0.1 were used for infection of THP-1-ASC-GFP cells. BMDMs were infected with a MOI of 2. For specific experiments, MOIs of 0.5, 1, 1.5, 5, and 10 were used to infect BMDMs. At different time points, cells or supernatants were recovered for RNA or protein analysis. Culture supernatants were filter sterilized prior to be used for protein detection. For immunofluorescent detection of ASC specks, infected cells were washed with PBS and fixed with formalin 2%. For RIF treatment condition, bacteria were treated with 500 μg mL$^{-1}$ of antibiotic for 18–24 h before been pelleted. Loss of viability was confirmed by plating treated bacteria on 7H11 agar plates.

**THP-1 ASC speck quantification.** Fixed cells were stained with DAPI and an IN CELL analyzer 2000 was used for plate image acquisition. Images were analyzed using Fiji[67].

**Chemical inhibitors or agonists.** For specific experiments, cells were treated with 0.1, 1, or 10 μM of the NLRP3 inhibitor, MCC950 (InvivoGen), or the caspase-1 inhibitor, VX-760 (InvivoGen); 10 or 20 μM of the cGAS inhibitor, RU.521 (InvivoGen); 25 μg mL$^{-1}$ of the K$^+$ channel inhibitor, Glybenclamide (InvivoGen); 10 μM of the cathepsin B inhibitor, ZRLR (kindly provided by Eva Wieczerzak[68]); 5 μM of the phagocytosis inhibitor, cytochalasin D (Sigma-Aldrich); or with 5 mM of ATP (InvivoGen); 1 μg mL$^{-1}$ of the RIG-I agonist, 5'ppp-dsRNA (InvivoGen).

**mRNA analysis by real-time PCR.** Total RNA from infected cells was extracted with TRIzol Reagent (Invitrogen, California, USA), according to the manufacturer's instructions. cDNA was synthesized using the ProtoScript First Strand cDNA Synthesis kit (New England Biolabs, Massachusetts, USA) and gene expression was analyzed by RT-PCR using the Bio-Rad CFX Manager™ 3.1, as describe before[69]. Targeted Il1β, Ifnβ, Tnf, and Il10 mRNA expression was quantified using SYBR Green (Thermo Fisher) and specific oligonucleotides, and normalized to ubiquitin mRNA levels. Ccl2 and Cxcl10 mRNA expression was quantified using TaqMan gene expression master mix (Applied Biosystems), and normalized to hypoxanthine phosphoribosyltransferase (HPRT). Oligonucleotide information is in Supplementary Table 5.

**Cytokine detection.** Cytokines were detected in supernatants of in vitro infected cultures by immunoassay. This included Multiplex (Invitrogen) for human IL-1β, IL-12p40, IL-10, and IFN-β (catalog numbers EPX01A-10224-901, EPX01A-12090-901, EPX01A-10215-901, and EPX01A-12088-901, respectively; all with the human basic kit, cat number EPX010-10420-901), or ELISA for mouse IL-1β (Invitrogen; BMS6002TEN), IFN-β (BioLegend; 439407) and TNF (Invitrogen; BMS607-3TEN) as indicated in the Figure legends.

**Global transcriptomic and pathway analyses.** Total RNA from infected or non-infected cells was extracted as above and treated with Turbo DNA-free kit (Invitrogen) before sequencing. Targeted RNA sequencing was performed by GenCore, i3S (Institute of Health Innovation and Research) using Ion AmpliSeq™ Transcriptome Mouse Gene Expression Kit. All subsequent analyses were performed in R [v3.5.1]. RNA expression levels in the count matrix were normalized using the trimmed mean of M-values and computed as counts per million using edgeR package [v3.24.1][70,71]. Differential expression analysis was performed using edgeR and limma [v3.38.3][72] packages. Genes with <15 raw counts in all samples were excluded. Differentially expressed genes were determined through linear model fitting. Empirical Bayes moderated t-statistics test was performed and genes with adjusted p-value ≤ 0.05 and log2 fold-change ≤ −2 or ≥2 were considered significant. Pathway analysis was performed using ReactomePA package [v1.26.0][73]. Adjustment of p-values for multiple testing was performed using the Benjamini–Hochberg (BH) procedure[74] in both differential expression analysis and pathway analysis.

**Quantification and statistical analysis.** Data were analyzed using GraphPad Prism software, version 8.1.0. Reference to "n" is in Figure legends. Unpaired, two-tailed Student's t-test was used to determine differences between two different groups and one-way ANOVA for more than two groups. Post-tests were applied to multiple comparisons as referred in Figure legends. Data was checked for normality and log normality. Other statistical tests included Fisher's exact test and Pearson correlation, as indicated in the Figure legends. Pearson's r was calculated to examine relationships between the concentration of ex vivo-induced cytokines (IL-1β, IL-12p40, or IL-10) and the TB presentation (mild, moderate, or severe) in the patients from which the bacteria were isolated. Differences were considered significant for p ≤ 0.05 and represented as follows: *p ≤ 0.05; **p ≤ 0.01; ***p ≤ 0.001 and ****p ≤ 0.0001. Bean plots were generated with the R package beanplot[75].

**Reporting summary.** Further information on research design is available in the Nature Research Reporting Summary linked to this article.

## Data availability

Sequence data are deposited on NCBI GEO database with the accession code GSE138580. All other data are available from the corresponding author upon reasonable request. Raw data for the figures are available as a Source Data file.

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

## Acknowledgements

This work was financed by FCT-Fundação para a Ciência e a Tecnologia/ Ministério da Ciência, Tecnologia e Inovação grant POCI-01-0145-FEDER-028955 (to M.S.) and by the Northern Portugal Regional Operational Program (NORTE 2020), under the Portugal 2020 Partnership Agreement, through the European Regional Development Fund (FEDER) (NORTE-01-0145-FEDER-000013, to M.I.V., F.R., A.G.C., and N.S.O.). I.C. acknowledges the support of Ministerio de Ciencia, Innovación y Universidades (SAF2016-77346-R) and the European Research Council (638553-TB-ACCELERATE). H.N.B. acknowledges the support of Bolsa D. Manuel de Mello and of the Portuguese Society for Pneumology; A.B. and M.S. were also recipients of an International Exchanges Grant from the Royal Society. J.S. is funded by a research fellow NORTE-01-0145-FEDER-000012; B.C. and K.L.F. are funded by FCT Ph.D scholarships SFRH/BD/114403/2016 and SFRH/BD/114405/2016, respectively; M.I.V. is funded by FCT through DL 57/2016 (CRP) and MS through Estimulo Individual ao Emprego Científico. We thank the excellent support from the i3S scientific platforms, namely Animal facility, Advanced Light Microscopy, and BioSciences Screening, member of the national infrastructure PPBI-Portuguese Platform of Bioimaging (PPBI-POCI-01-0145-FEDER-022122) and Genomics (GenCore) part of the GenomePT project (POCI-01-0145-FEDER-022184), supported by COMPETE 2020 - Operational Programme for Competitiveness and Internationalisation (POCI), Lisboa Portugal Regional Operational Programme (Lisboa2020), Algarve Portugal Regional Operational Programme (CRESC Algarve2020), under the PORTUGAL 2020 Partnership Agreement, through the European Regional Development Fund (ERDF), and by Fundação para a Ciência e a Tecnologia (FCT).

## Author contributions

Conceptualization: J.S., B.C., F.R., P.R., S.G., A.G.C., J.T.G., H.N.B., N.S.O., I.C., M.S. Formal analysis: J.S., B.C., A.R.M., L.S.C., C.M., A.O.C., J.V., C.V., H.N.B., N.S.O., I.C.,

M.S. Funding acquisition: A.B., H.N.B., I.C., M.S. Investigation: J.S., B.C., A.R.M., K.L.F., A.I.F., L.B., H.M., M.I.V., A.S., R.P., A.A., A.B., H.N.B. Project administration: M.S. Resources: A.R., T.C., R.P., A.A., J.T.G. Software: A.R.M., L.S.C., C.M., A.O.C., N.S.O., I.C. Supervision: S.G., J.T.G., N.S.O., I.C., M.S. Visualization: J.S., B.C., C.M., H.N.B., N.S. O., I.C., M.S. Writing-original draft: J.S., S.G., A.G.C., H.N.B., N.S.O., I.C., M.S. Writing-review and editing: all authors.

## Competing interests

The authors declare no competing interests.
