## [Peer Review File · Nature Communications]

Reviewers' comments:

Reviewer #1 (Remarks to the Author):

This is a comprehensive and informative manuscript about the complexities of TB infections and how some TB strains evade and avoid host defense systems. Importantly, the authors begin with many patients and strains; necessarily, they focus on a narrower subset to perform their more mechanistic studies. There are some elements of the manuscript that can be improved:

1. I found the summary somewhat vague. For example, IL1b is not mentioned. In a way, the authors undersold their approach and data in the summary.
2. IGRA needs to be defined (including its importance to the study) for the general reader.
3. The biggest issue is the presentation of the data in Figures 1 and 2. 1a,b are not informative. Especially confusing is 2a, b (2c, d could also be improved). For example, in 2a it is entirely unclear what is being shown. Surely there are better ways to display the data. 2b should be a conventional graph.
4. In 5a and 6a, the data should be shown as z-scores rather than absolute expression

Reviewer #2 (Remarks to the Author):

The study is very interesting. The authors show in a very elegant way that the clinical outcome of TB is due to decreased IL-1 β responses which are caused by a specific M. tuberculosis.

Few revisions are needed:

1. INTRODUCTION:

a. TB data worldwide: please, report the updated data as in the WHO reference of 2019 and report the reference

2. RESULTS

a. row 113. Sentence incomplete " against cells from active TB patients, as their cytokine responses are altered by the disease status"

3. DISCUSSION

a. discuss the results in light of the IL-1 modulation and correlation with cell autophagy, as reported in Romagnoli et al, Cell Death Dis. 2018

4. METHODS:

a. Clinical severity classification: please, describe better the radiology criteria to define the status of the patients as "severe, mild, medium"

Reviewer #3 (Remarks to the Author):

The authors of the manuscript "Mycobacterium tuberculosis associated with severe tuberculosis evades cytosolic surveillance systems and modulates IL-1 β production" present a study where they have studied 618 patient isolates and have tried to establish a correlation between severity of TB cases, the capacity of inducing weak or strong types of cytokine responses in macrophages and genotype of the M. tuberculosis isolates. The authors conclude that mild TB cases were caused by bacteria that induced strong cytokine responses in macrophages while the causative strains of severe TB cases induced weak activation of the cytosolic surveillance systems and weak cytokine responses in macrophages. The authors have selected representative strains of each group (within the lineage 4 group of Euro-American strains), with which they have conducted certain cell biological experiments to try to link these features to mutations leading to sigA recognition boxes or affecting components of the ESX-1 secretion system.

The authors took IL-1 β as a proxy for low versus high cytokine responses. Analysis of SNPs selected 5 SNP positions from their Porto dataset also emerged in unrelated strains in the

phylogeny more than one time. As one example, a one of the synonymous mutations among the 5 SNPs generated a new internal TANNT sigA recognition box, which is described as a putative mechanism for *M. tuberculosis* to adapt to changing environments. Other such SNPs were found in genes associated to the ESX-1 type VII secretion system.

In general, the article provides interesting hypotheses about the association between *M. Tuberculosis* genotypes and severity of TB disease. The authors also provide a dataset to underline their arguments. However, at several occasions the direct link between genotype and phenotype seems weak and it is not clear whether the declared association might relate to other causes than those suggested by the authors.

For example, in lane 153 the authors claim that they had found common genetic variants shared by 5 clinical isolates (6C4, 3A7, 1C7, 5F9 and 5D6) that were inducing low IL-1 β levels, whilst absent in high IL-1 β inducers (Figure 2D; Table S3), and several of these were associated with genes encoding ESX-1 (associated) proteins. However, there was only little follow-up of this observation and the respective strains were not tested to see if their level of ESAT-6 secretion was modified. According to the literature, ESAT-6 is one of the main proteins required for induction of phagosomal perforation and induction of cytosolic signaling events that lead to enhanced IL1-beta secretion. However, mutations in ESX-1 are normally related to low virulence associated with low cytosolic signaling.

Thus, it would be important for the study to measure the level of ESAT-6 secretion and the potential of the concerned strains to induced phagosomal damage. In previous work some of the authors of the paper have demonstrated a link between inflammasome activation and ESAT-6 secretion. For the moment it is not clear how mutations upstream of EspR might lead to low IL-1beta induction and in the same time to higher virulent strains that can induce severe TB.

The same applies to the data on cGAS induction, while the authors measure the induction of cGAS, the bacterial factors that are usually associated with the phenotype were not tested (such as ESAT-6 secretion and the ability to induce phagosomal damage).

Finally, it would be very helpful for claiming a link between a particular SNP with a particular IL-1beta induction phenotype if the authors could demonstrate an association by using genetic complementation strategies. It might have been most useful to introduce the WT allele of the mutated genetic region that the authors suspect to be involved in low IL-1 beta induction into the clinical isolate they have worked with in their experimental part of the work, and then test the complemented strains for the IL-1 beta induction phenotype. If the association is real, this should also be reflected by the phenotype of the complemented strains.

In general, many experiments used downstream techniques to show that certain *M. tuberculosis* strains have a different capacity to induce certain levels of immune signaling and IL-1 beta responses, but there are only few experiments described in the manuscript that focus actively on the bacterial factors that might be involved. While the hypotheses are attractive, the experimental demonstration often lacks clarity and needs updating by the authors in order to stronger support the claims made by the authors between certain SNPs, IL-1beta responses and severity of TB.

Reviewers' comments:

Reviewer #1 (Remarks to the Author):

This is a comprehensive and informative manuscript about the complexities of TB infections and how some TB strains evade and avoid host defense systems. Importantly, the authors begin with many patients and strains; necessarily, they focus on a narrower subset to perform their more mechanistic studies.

We thank the reviewer for these encouraging comments and for appreciating our strategy. We also thank the suggestions provided, which we have now addressed as described below. All changes are highlighted in yellow in the revised manuscript.

There are some elements of the manuscript that can be improved:

1. I found the summary somewhat vague. For example, IL1b is not mentioned. In a way, the authors undersold their approach and data in the summary.

We appreciate this point and have now improved the abstract to make it more reflective of our approach and data. Please see page 2, revised manuscript.

2. IGRA needs to be defined (including its importance to the study) for the general reader.

We thank the reviewer for raising this point and have now improved our description on this topic. Please see page 4, revised manuscript.

3. The biggest issue is the presentation of the data in Figures 1 and 2. 1a,b are not informative. Especially confusing is 2a, b (2c, d could also be improved). For example, in 2a it is entirely unclear what is being shown. Surely there are better ways to display the data. 2b should be a conventional graph.

We understand the reviewer's point. Although the data in Figure 1a could indeed be referred in the text only, we think it makes it easier to the reader to provide the distribution of the comorbidities present in the cohort in the context of Figure 1, so would suggest to maintain as is.

Figure 1b refers to the ancestry analysis of the patients in our cohort, which is interesting to show in parallel with the bacteria ancestry analysis. We would like to maintain this panel, but agree with the reviewer that it is not very informative as is. We have now added to this panel the ancestry analysis of the control populations (which was shown as supplementary data in the original manuscript), thus making it easier for the reader to relate the cohort under study, not only with other Portuguese populations, but also with global ones. Please see page 3 and 29 and Figure 1, revised manuscript.

Figure 2a is indeed not well explained and in fact not needed, as it shows is the pairwise distance between the selected isolates. This is now mentioned in the text only. Please see pages 5-6 and 29-30 and Figure 2, revised manuscript.

Following the reviewers Figure 2b (now 2a) was converted in a conventional phylogenetic tree. We also tried to improve the colour scheme of Figures 2c and 2d (now 2b and c). Please see Figure 2, revised manuscript.

4. In 5a and 6a, the data should be shown as z-scores rather than absolute expression.

These figures are now presented as z-scores. Our rational to not have done so, was to compare relative gene expression levels across different genes, as well as for the same gene across the different groups. We agree with the reviewer and this is now modified. For consistency, we have also altered Supp. Figures 4b, 5g, and 6a, b and c. Please see Figure 5a and 6a, Supp. Figures and pages 30-31, revised manuscript.

Reviewer #2 (Remarks to the Author):

The study is very interesting. The authors show in a very elegant way that the clinical outcome of TB is due to decreased IL-1 β responses which are caused by a specific M. tuberculosis.

We thank the reviewer for these encouraging comments. We also thank the suggestions provided, which we have now addressed as described below. All changes are highlighted in yellow in the revised manuscript.

Few revisions are needed:

1. INTRODUCTION:

a. TB data worldwide: please, report the updated data as in the WHO reference of 2019 and report the reference

Done. Please see pages 2 and 21, revised manuscript.

2. RESULTS

a. row 113. Sentence incomplete “ against cells from active TB patients, as their cytokine responses are altered by the disease status”

Corrected. Please see page 4, revised manuscript.

3. DISCUSSION

a. discuss the results in light of the IL-1 modulation and correlation with cell autophagy, as reported in Romagnoli et al, Cell Death Dis. 2018

This is now included. Please see page 12, revised manuscript.

4. METHODS:

a. Clinical severity classification: please, describe better the radiology criteria to define the status of the patients as “severe, mild, medium”

We have now improved this. Please see page 14, revised manuscript.

Reviewer #3 (Remarks to the Author):

The authors of the manuscript “Mycobacterium tuberculosis associated with severe tuberculosis evades cytosolic surveillance systems and modulates IL-1 β production” present a study where they have studied 618 patient isolates and have tried to establish a correlation between severity of TB cases, the capacity of inducing weak or strong types of cytokine responses in macrophages and genotype of the M. tuberculosis isolates. The authors conclude that mild TB cases were caused by bacteria that induced strong cytokine responses in macrophages while the causative strains of severe TB cases induced weak activation of the cytosolic surveillance systems and weak cytokine responses in macrophages. The authors have selected representative strains of each group (within the lineage 4 group of Euro-American strains), with which they have conducted certain cell biological experiments to try to link these features to mutations leading to sigA recognition boxes or affecting components of the ESX-1 secretion system.

The authors took IL-1 β as a proxy for low versus high cytokine responses. Analysis of SNPs selected 5 SNP positions from their Porto dataset also emerged in unrelated strains in the phylogeny more than one time. As one example, a one of the synonymous mutations among the 5 SNPs generated a new internal TANNT sigA recognition box, which is described as a putative mechanism for M. tuberculosis to adapt to changing environments. Other such SNPs were found in genes associated to the ESX-1 type VII secretion system.

In general, the article provides interesting hypotheses about the association between M. Tuberculosis genotypes and severity of TB disease. The authors also provide a dataset to underline their arguments. However, at several occasions the direct link between genotype and phenotype seems weak and it is not clear whether the declared association might relate to other causes than those suggested by the authors.

For example, in lane 153 the authors claim that they had found common genetic variants shared by 5 clinical isolates (6C4, 3A7, 1C7, 5F9 and 5D6) that were inducing low IL-1 β levels, whilst absent in high IL-1 β inducers (Figure 2D; Table S3), and several of these were associated with genes encoding ESX-1 (associated) proteins. However, there was only little follow-up of this observation and the respective strains were not tested to see if their level of ESAT-6 secretion was modified. According to the literature, ESAT-6 is one of the main proteins required for induction of phagosomal perforation and induction of cytosolic signaling events that lead to enhanced IL-1-beta secretion. However, mutations in ESX-1 are normally related to low virulence associated with low cytosolic signaling.

Thus, it would be important for the study to measure the level of ESAT-6 secretion and the potential of the concerned strains to induced phagosomal damage. In previous work some of the authors of the paper have demonstrated a link between inflammasome activation and ESAT-6 secretion. For the moment it is not clear how mutations upstream of EspR might lead to low IL-1beta induction and in the same time to higher virulent strains that can induce severe TB.

The same applies to the data on cGAS induction, while the authors measure the induction of cGAS, the bacterial factors that are usually associated with the phenotype were not tested (such as ESAT-6 secretion and the ability to induce phagosomal damage).

Finally, it would be very helpful for claiming a link between a particular SNP with a particular IL-1beta induction phenotype if the authors could demonstrate an association by using genetic complementation strategies. It might have been most useful to introduce the WT allele of the mutated genetic region that the authors suspect to be involved in low IL-1 beta induction into the clinical isolate they have worked with in their experimental part of the work, and then test the complemented strains for the IL-1 beta induction phenotype. If the association is real, this should also be reflected by the phenotype of the complemented strains.

In general, many experiments used downstream techniques to show that certain M. tuberculosis strains have a different capacity to induce certain levels of immune signaling and IL-1 beta responses, but there are only few experiments described in the manuscript that focus actively on the bacterial factors that might be involved. While the hypotheses are attractive, the experimental demonstration often lacks clarity and needs updating by the authors in order to stronger support the claims made by the authors between certain SNPs, IL-1beta responses and severity of TB.

We thank the reviewer for these comments and for the insightful suggestions on our work.

The reviewer is absolutely right in pointing out that our study is focused in the downstream mechanisms induced by different M. tuberculosis strains in host macrophages. We indeed investigated in higher detail which and how host pathways are exploited by the bacteria to manipulate the macrophage cytokine response and ultimately how that articulates with

differential TB presentations. We also agree that there are still many open and very interesting questions on the bacteria side, which we are pursuing in the lab.

We are gathering information on other mild and severe/ high or low IL-1b-inducing isolates, to increase our datasets and restrict SNP candidates. Our findings (presented in the manuscript) highlight that several SNPs and/or combination of SNPs are likely to contribute to the phenotype. Furthermore, as we show, the ability of certain M. tuberculosis isolates to limit IL-1b production does not result from ancestral events. Therefore, as different recent genetic «solutions» may explain the phenotype, restricting the candidates is important to guide our genetic approach, where (as suggested by the reviewer) we plan to construct genetically modified strains to finally probe the association of candidate SNPs with the respective phenotype. In this context, we will also tackle the quantification of ESAT6 secretion and of the phagolysosome rupture. In particular, we plan to address these questions in infected macrophages and over time. The kinetic aspect is particularly important, because the strains under study may be manipulating phagolysosome damage to fine tune it, not necessarily to abrogate it. As the reviewer points out, “mutations in ESX-1 are normally related to low virulence associated with low cytosolic signalling”, so our hypothesis is that virulent strains (ie those causing disease in humans, as all our isolates) might not have abrogated this pathway, but instead fine-tuned it. We are creating the experimental setup needed to pursue our study on the bacteria side, which is per se a medium-term project. Given the fact that our study uncovers a novel link between M. tuberculosis diversity, manipulation of the immune response and TB manifestation, and provides mechanistic insights on how this link operates in the host, we consider that fully developing the bacteria-associated determinants and mechanisms is the scope of another study.

In this context, and taking into consideration the relevance of the reviewer’s comments, we have extended the discussion of our manuscript, to clearly state the interesting points to be pursued on the pathogen side following up the hypotheses raised by our in silico approach. Please see page 11, revised manuscript.

Moreover, if the reviewer finds it of interest, we are prepared to move the findings from Figure 2 into a final supplementary figure presented in the context of the discussion and in this way clearly spell out the in silico data as an initial approach leading to interesting hypothesis that are not addressed in this manuscript.